# Lymphopenia Caused by Virus Infections and the Mechanisms Beyond

**DOI:** 10.3390/v13091876

**Published:** 2021-09-20

**Authors:** Zijing Guo, Zhidong Zhang, Meera Prajapati, Yanmin Li

**Affiliations:** 1State Key Laboratory on Veterinary Etiological Biology, Lanzhou Veterinary Research Institute, Chinese Academy of Agricultural Sciences, Lanzhou 730030, China; zijingguo7@163.com; 2College of Animal Husbandry and Veterinary Medicine, Southwest Minzu University, Chengdu 610041, China; zhangzhidong@swun.edu.cn (Z.Z.); prajapati_m@hotmail.com (M.P.); 3National Animal Health Research Centre, Nepal Agricultural Research Council, Lalitpur 44700, Nepal

**Keywords:** lymphopenia, viruses, apoptosis, cytokines, immune response

## Abstract

Viral infections can give rise to a systemic decrease in the total number of lymphocytes in the blood, referred to as lymphopenia. Lymphopenia may affect the host adaptive immune responses and impact the clinical course of acute viral infections. Detailed knowledge on how viruses induce lymphopenia would provide valuable information into the pathogenesis of viral infections and potential therapeutic targeting. In this review, the current progress of viruses-induced lymphopenia is summarized and the potential mechanisms and factors involved are discussed.

## 1. Introduction

Lymphocytes are important elements of the immune system. They are categorized into T lymphocytes (T cells), B lymphocytes (B cells) and natural killer (NK) cells based on their migration, surface makers and biological functions [1]. Lymphopenia is the condition in which there is an abnormal reduced number of lymphocytes in the peripheral blood. It is diagnosed when a total lymphocyte number is lower than normal for a particular age group (for instance, less than 1000 cells/μL in older children and adults) [2,3,4]. Such obvious reduction in blood lymphocytes count occurs due to viral infections [5], chemical and physical lympho-depleting agents [6], autoimmune-related systemic diseases [7], genetic factors [8], cancers [9], sepsis [10] and other severe injuries [11]. Generally, most viruses lead to relative lymphocytosis, while only a few viruses causing severe disease could result in lymphopenia, such as severe acute respiratory syndrome coronavirus-2 (SARS-CoV-2) [12], ebola virus (EBOV) [13] and human immunodeficiency virus (HIV) [14]. Although numerous studies were attempted to reveal the causes of lymphopenia during viral infections, the mechanisms underlying it is still unclear. It is believed that the mechanism underlying lymphocyte depletion is more complicated as different factors or mechanisms are involved during the infection caused by different viruses. In this review, the current progress of the lymphopenia induced by viral infections with emphasis on RNA viruses is summarized and the mechanism and factors involved during various virus infection-mediated lymphopenia are discussed.

## 2. Mechanisms of Lymphopenia Associated with Viral Infections

Lymphopenia was observed in patients and animals infected with different viruses which belong to different viral families and thus exhibit varied morphology, physicochemical and physical properties, genome organization and replication and antigenicity. However, infections by most of these viruses could cause serious illness or even death and it was found that lymphopenia is associated with disease severity. Despite numerous studies on causes of lymphopenia during viral infections, the mechanisms underlying still remain vague. The proposed mechanisms for the observed lymphopenia are summarized as follows (Table 1, Figure 1). Unlike the RNA viruses mentioned below, the mechanism of lymphopenia caused by DNA virus infections is also still unknown. So far, the underlying factors and molecular mechanisms of lymphopenia caused by most DNA viruses have rarely been reported which needs to be addressed in the future.

### 2.1. Cell Death

Lymphocyte death plays a critical role in lymphopenia caused by many viral infections (Table 1). Apoptosis, pyroptosis, autophagy, virus-specific CD8+ cytotoxic T lymphocyte (CTL) dependent killing and antibody-dependent cell-mediated cytotoxicity (ADCC) were reported to be involved in lymphopenia. Among them, apoptosis was thought to be a major pathway involved in lymphopenia [39,62,68].

Apoptosis: Apoptosis is a tightly regulated form of cell death that is vital in both embryo implantation and development and turnover of tissues during maturation. Lymphocyte’s apoptosis has been widely reported during viral infections, such as SARS-CoV-2 [15], HIV [39] and IAV [62]. Until now, the mechanism responsible for lymphocytes apoptosis is not fully understood. Although apoptosis of lymphocytes could be induced due to direct infection of lymphocytes with the viruses, such as MERS-CoV [23,24], HIV [111] and MV [67], the number of such virus-infected cells was very few to match with the reduced levels of lymphocytes observed during viral infections. Some studies even suggested that lymphocytes could not be infected by other viruses, such as SARS-CoV-2 [15] and FMDV serotype O/A [81,84]. Thus, apoptosis of bystander cells is considered as a major mechanism for lymphocytes depletion induced by viral infections [39,112,113]. There were many factors reported to trigger apoptosis in uninfected bystander lymphocytes, including the activation of apoptosis related-receptors (Fas/FasL) [61], the interaction between viral protein and host cellular receptors [114] and cytokines (TNF-α) [115,116].

Activation-induced cell death (AICD): The interaction between FasL and Fas is usually engaged in AICD, which enhances apoptosis in T lymphocytes previously stimulated to cause contraction of T cells when T lymphocytes are re-activated via T cell receptors (TCR) [117,118,119]. AICD in peripheral T cells is often caused by the induction of expression of the death ligand, Fas ligand (CD95 ligand, FasL). During SARS-CoV-2-, RSV- and CDV- infection, the Fas expression was negatively correlated to CD4+ T lymphocyte count in blood, indicating that the increased expression of Fas was involved in lymphopenia through apoptosis of bystander T cells [70,71,76,119].

Dendritic cell (DC)-dependent killing of lymphocytes: Previous studies showed that a depletion of CD8+ T cell responses in lethal H2N2 influenza virus infection was mediated by lymph node resident DCs, especially plasmacytoid dendritic cells (pDCs) that express FasL and drive FasL–Fas induced T cell apoptosis in a dose-dependent manner [120,121]. Similarly, H5N1 infection in mice enhanced FasL expression on pDCs, resulting in apoptosis of influenza-specific CD8+ T cells via a Fas–FasL-mediated pathway [61,121]. In addition, HIV-infected DCs induced CD8+ T cell apoptosis by up-regulating TNF-α in infected DCs and activation of the caspase 8-dependent pathway in CD8+ T cells [122].

Pyroptosis: Pyroptosis is a programmed cell death with high inflammation. The dying cell releases its cytoplasmic contents, including inflammatory cytokines [123]. These cytokines in turn induce pyroptosis in other T cells, which also contribute to T cells depletion. IL-1β, a marker of pyroptosis, was increased during SARS-CoV-2 and HIV infection, indicating that lymphocytes were undergoing pyroptosis. During CSFV infection, pyroptosis was also determined in peripheral lymphoid organs by TdT-mediated dUTP nick-end labeling (TUNEL) and detection of pyroptosis related genes (such as p10 subunit, caspase-1 and IL-1β) [87]. It is necessary to further investigate how these viral infections induce lymphocytes pyroptosis.

Autophagy: Autophagy is an important component of anti-viral defense in host cells. During autophagy, the viruses are initially pinned down into autophagosomes and then delivered to lysosomes for degradation [124]. Until now, autophagy was only reported in patients infected with SARS-CoV-2, HIV and EBOV. In COVID-19 patients, the upregulated expressions of autophagy-associated genes have been observed in PBMCs, which indicates that these cells may undergo autophagy and ultimately cell death [15]. In HIV infected patients, autophagy mediated by HIV gp41 protein was reported in the uninfected CD4+ T cells [41]. In EVD patients, autophagy was induced in T cells through activating endoplasmic reticulum (ER) stress related signaling pathways [13].

Virus-specific CD8+CTL dependent killing: The CTL remove the virus-infected lymphocytes through interaction of FasL with Fas and TNF-related apoptosis-inducing ligand (TRAIL–TRAILR). CTL induce apoptosis of targeted cells by perforin and granzymes and by the death ligand, CD95 ligand (CD95L) [125]. Although a small proportion of lymphocytes infected with SARS-CoV-2 could be eliminated by CTL-dependent killing [126], it is believed that CTLs are not likely to be involved in lymphocytes death due to the lack of infection. Thus, CD8+CTL dependent killing is more likely to occur in lymphocytes infected with MERS-CoV, HIV, CDV, etc.

ADCC: Antibodies specific to the viral surface antigens are able to block attachment of a virus to its target cell. However, virus-infected cells could be also bound by the antibodies against the virus by recognition of viral antigens presented on the cells. There are a variety of mechanisms involved in killing the antibody-coated cells infected with the virus, such as phagocytosis, activation of complements and ADCC [127]. It is noted that more severe lymphopenia and higher titers of IgG and IgM against the virus were observed in patients, who had severe COVID-19 in comparison with patients having moderate and mild severity [16].

### 2.2. Involvement of Cytokines, Chemokines and Growth Factors

Upregulated expression of cytokines, chemokines and growth factors was commonly observed in patients and animals with lymphopenia induced by viral infection, including IL-2, IL-6, IL-12, IL-18, IL-1β, IFN-γ, CCL2/MCP-1, CXCL1, CXCL8/IL-8, CCL3/MIP-1-α, CCL7/MCP-3, CXCL10/IP-10 and CXCL9/MIG [17,53,128]. It is noted that the kinds of elevated cytokines, chemokines and growth factors caused by varied viral infections is different, i.e., the infections of SARS-CoV, MERS-CoV, HIV and FMDV could trigger higher levels of IFNs. Conversely, the infection of SARS-CoV-2 did not induce any IFN expression at all assessment time points [129]. Among cytokines, chemokines and growth factors reported, IL-6 is the most common cytokine which was upregulated during infections with many viruses including SARS-CoV-2, SARS-CoV, HIV, EBOV, IAV, RSV, CSFV and HNoV. Different cytokines induced lymphopenia through different mechanisms; it is believed that the synergic action of cytokines, chemokines and growth factors plays a vital role in induction of lymphopenia during viral infection, although the molecular mechanism beyond remains unclear.

IL-6, produced rapidly in response to infection and tissue damage, promotes host defense by stimulating the acute phase responses or immune response. However, the excessive release of IL-6 plays a pathological role in chronic infection and inflammation [130]. Notably, IL-6 could inhibit lymphopoiesis by directly impairing hematopoietic stem cells [131] and STAT-3 activation [132]. Recent studies found that T cells counts were negatively correlated to IL-6, IL-10 and TNF-α concentration in serum, which indicated that these cytokines may be involved in T cells depletion in COVID-19 patients [18,19]. The drugs blocking IL-6 receptor (IL-6R), such as tocilizumab, sarilumab and siltuximab, have been considered as a treatment strategy for severe COVID-19 patients with high IL-6 levels [133]. Importantly, lymphocytes count was recovered after treatment with IL-6R [134].

IL-10 as an anti-inflammatory cytokine exhibits a dual role in promoting pathogen persistence [135] and limiting immune pathology [136]. A previous study found that IL-10 was able to suppress proliferation of T cells [137]. Study on chronic infection in animal models showed that blocking IL-10 signaling could successfully prevent T cell exhaustion [138]. In addition, T cell apoptosis could be induced by IL-10 produced by CD9+ regulatory B cells.

TNF-α is also a pro-inflammatory cytokine and could induce cell apoptosis through interacting with its receptor TNFR1 [115]. Previous studies suggested that TNF-α produced by macrophages infected with ASFV leads to apoptosis in bystander T lymphocytes [116]. Similarly, apoptosis was observed earlier in lymphocytes from FIP cats [34], and apoptosis was further reported to be caused by TNF-α or other potential cytokines, instead of viral direct infection [33,35].

Interferons (IFNs) were originally identified as a humoral factor that confer an antiviral state on cells [139], but now, it has also been found to regulate lymphocyte recirculation and cause transient blood lymphopenia [140]. During IAV, CSFV and FMDV infection, the onset of lymphopenia was consistent with the IFN-α responses, and all animals with a high IFN-α level in serum showed severe depletion of lymphocytes [63,85,88]. IFN-γ has a critical part in bridging the innate and adaptive immunity, but IFN-γ has been reported to promote apoptosis. It has been proposed that IFN-γ may promote apoptosis of the bystander T cell following EBOV infection [141]

In addition to IL-6, IL-10, TNF-α and IFNs, other cytokines, chemokines and growth factors were also involved in lymphopenia. For instance, IL-15 has been reported to affect proliferation and differentiation of lymphocytes [142], MCP-1 and CXCL13 were reported to be involved in the migration of lymphocytes [42,143,144].

### 2.3. Inhibition of Lymphopoiesis

There are many ways to inhibit lymphopoiesis during viral infection. First, the damage of lymphoid organs (thymus, lymph nodes, bone marrow, liver and lung) was observed in severe illness caused by SARS-CoV-2, CDV and CSFV with lymphopenia, which has negative impacts on the survival, production and function of lymphocytes [87,89,145,146,147]. The thymus, an important lymphoid organ, is responsible for the generation and maturation of T cells. However, thymic dysfunction and involution have been observed in patients infected with HIV [148], SARS-CoV-2 [149], MV [150] and H1N1 [151], which were triggered by direct thymocyte killing. Thymus suppression affects lymphopoiesis and the survival of lymphocytes. Second, the development of haematopoietic precursor cells (HPC) was suppressed through viral direct infection and the synergic action of cytokines, chemokines and growth factors. Previous studies postulated that SARS-CoV-2 and CSFV may directly infect haematopoietic stem cells (HSC) and thus induce cell death [20,90]. In addition, hyperproinflammatory cytokines are produced by activated immune cells that affect the process of lymphopoiesis in the bone marrow. Studies have suggested that the reciprocal dynamics of lymphocyte and neutrophil populations in the bone marrow are consistent with cellular interaction and competition [152]. However, inflammation could regulate the balance of granulopoiesis and lymphopoiesis in bone marrow by suppressing common proinflammatory cytokines and growth factors (TNF-α, IL-1β, CXCL12, IL-6, etc) that affect lymphopoiesis more strongly than granulopoiesis [152,153]. TNF-α treatment results in a reduction in lymphocyte progenitor populations in the bone marrow, while IL-1β elicits increased granulocyte precursors [154]. Moreover, CXCR4-CXCL12 interactions facilitate cytokine-mediated regulation of B cell and myeloid cell retention in the bone marrow [154]. IL-6 has been reported to abort lymphopoiesis and elevate production of myeloid cells by expression of Id1 transcription factor, which is known to inhibit lymphopoiesis and elevate myelopoiesis, and its expression was dependent on MAPK [131]. Although the cellular source of these cytokines was not determined during many virus infections (e.g., the cellular source of elevated IL-6 in COVID-19 patients [155]), the synergic action of these proinflammatory cytokines may inhibit lymphopoiesis in the bone marrow. Lastly, inhibition of the lymphocyte activation and proliferation by the loss of the antigen-presenting ability would also promote lymphopenia associated with viral infections. For example, DCs from healthy and convalescent COVID-19 patients could stimulate T cell proliferation, but none of the DCs derived from acute COVID-19 patients showed similar activity, which indicated that DCs from acute patients showed functional impairment in both maturation and T cell activation [21]. Similarly, DCs infected with EBOV and IAV inhibit the ability of antigen presentation and affect the capacity of activating T cells thereby limiting the ability to pass survival signals to T cells [54,64,156]. Further studies revealed that the activation of T cells was inhibited due to steric shielding, in which EBOV GP protein expressed on infected antigen presenting cells (APCs) masks its own epitopes and MHC-I and β1 integrin expressed on the cell surface [55]. Furthermore, TCR signal transduction could be inhibited via IFN-inhibiting domains located in viral VP35 and VP24 proteins when EBOV-infected DCs were in contact with T cells [157,158].

### 2.4. Lymphocyte Trafficking

Accumulation of lymphocytes was often observed in infected tissue sites of patients and animals, which was believed to be due to migration from peripheral blood [65,97]. The recovering SARS patients had an average increase of 121 CD4+ T cells per microliter of peripheral blood during the first month of disease onset; it is speculated that the rapid recovery in lymphocytes blood count is more likely due to the recirculation of lymphocytes between organs and peripheral blood, rather than newly produced lymphocytes [28,69]. In addition, various kinds of leucocytes infiltrate the alveoli at various degrees after SARS-CoV-2- and IAV- infections, including lymphocytes, monocytes, neutrophils and eosinophils [65,159]. Therefore, lymphocyte sequestration in lungs is considered as a potential pathway for the depletion of blood lymphocytes [160]. Similarly, the loss of circulating lymphocytes temporally coincides with the accumulation of lymphocytes in the lymph nodes and jejunal mucosa following HIV-, SIV- and HNoV-infection, which suggests that lymphopenia may occur as a result of the redistribution of circulating lymphocytes to the infected sites [50,97,161]. Although the molecular mechanism for lymphocytes trafficking remains unclear, it seems that CXCL13, IP10 and IFN-α/β play an important role in lymphocyte trafficking [42,44,140].

### 2.5. Role of Co-Inhibitory Molecules

T cell activation requires two signals: TCR stimulation through antigen recognition in the context of MHC and co-stimulation through interaction of co-signaling receptors (co-stimulatory and co-inhibitory receptors) on T cells with their ligands on APCs [162]. Co-signaling receptors play a pivotal role in regulating T cell activation, subset differentiation, effector function and survival [162]. Co-inhibitory molecules include cytotoxic T-lymphocyte-associated protein 4 (CTLA-4), programmed death 1 (PD-1), T cell immunoreceptor with Ig and ITIM domains (TIGIT), lymphocyte activation gene-3 (LAG-3), T cell immunoglobulin mucin 3 (Tim-3) and 2B4. Up-regulated expression of these co-inhibitory molecules has been reported following viral infection, such as SARS-CoV-2 [22], HIV [45], EBOV [56], IAV [66] and RSV [77]. It is noted that the expression of PD-1, LAG-3 and TIGIT in CD4+ T cells showed the strongest inverse associations with the number of CD4+ T cell from AIDS patients [45]. During high-pathological IAV infection, the expression of PD-1 on CD8+ T cells specific for IAV was upregulated, and blockade of PD-L1 in vivo caused reduced titers of virus and increased numbers of CD8+ T cell numbers [66]. PD-1 has been reported to promote T cells exhaustion and apoptosis and to inhibit proliferation, cytokines production and cytolytic function by regulating signaling of AKT, phosphoinositide 3-kinase (PI3K), extracellular-signal regulated kinase (ERK) and phosphoinositide phospholipase C-γ (PLCγ) [163]. A recent study demonstrated that PD-1-mediated PI3K/Akt/mTOR, caspase9/caspase3 and ERK pathways are involved in regulating the apoptosis and proliferation of CD4+ and CD8+ T cells during BVDV infection. The molecular mechanism of other co-inhibitory molecules in lymphopenia caused by viral infection needs further study.

### 2.6. Metabolic Disorders

Lactic acid environments in tumors have been confirmed to suppress the proliferation and cytokines production of human CTLs and lead to a 50% decrease in cytotoxic activity, and the CTLs function can be restored in a lactic acid-free medium [164]. The level of lactic acid was significantly upregulated in severe COVID-19 patients compared to mild patients [165]. It is speculated that the elevated blood lactic acid levels in severe COVID-19 patients might suppress the proliferation of lymphocytes. In addition, the bilirubin levels (the marker of liver damage), urea nitrogen and creatinine (the markers of renal function) were significantly increased in severe COVID-19 patients [5,166,167]. The damage of these organs, especially liver and kidney, may affect the survival, production and function of lymphocytes. Similarly, RSV infected patients who required ICU showed a higher level of prolactin and growth hormone, while the leptin and insulin-like growth factor 1 (IGF1) were obviously decreased [78]. Further analysis revealed that the levels of prolactin and leptin were related to lymphocyte counts, which were considered as potential factors related to lymphopenia in severe RSV infection [78]. Overall, metabolic disorders might shift the normal physiological condition into a pathophysiological situation that may affect the production, survival and function of lymphocytes.

### 2.7. Glucocorticoids

Glucocorticoids not only exhibit anti-inflammatory actions but also trigger lymphocytes apoptosis that further contributes to lymphopenia [168,169]. Previous studies suggested that lymphopenia was more prevalent in SARS patients with higher prevailing cortisol, before any steroid therapy had been used [170]. Similarly, the blood cortisol was significantly higher during the active phase than during the convalescent phase after RSV- and EBOV-infection [78,171]. These phenomena revealed that glucocorticoids may be involved in lymphopenia. The role of glucocorticoids in lymphopenia has been reviewed by Panesar N.S. [31]. A recent review study described that the ordering between lymphopenia and lymphocytes apoptosis appears different in SARS and COVID-19 patients (apoptosis is prior to lymphopenia in COVID-19 patients), and the level of glucocorticoids could determine the ordering between lymphopenia [32]. Therefore, glucocorticoids appear to play an important role in lymphopenia caused by virus infections, and glucocorticoids as a drug should be used carefully for treatment in patients with lymphopenia.

## 3. Clinical Implications of Lymphopenia during Viral Infections

### 3.1. Association of Lymphopenia with Viral Disease Severity

Several studies have shown that severe and critically ill individuals during viral infection (such as SARS-CoV-2 or EBOV or AIV), who required intensive care unit (ICU), had obviously lower lymphocyte counts compared to those in healthy, mild or recovered individuals [13,172,173,174]. This phenomenon indicated that the degree of lymphocyte count has been associated with disease severity [174,175,176]. T cells play a critical role in virus-specific, adaptive immune response, thereby suggesting that lymphopenia could severely impair the body’s ability to fight infection. Likewise, Xu et al. also pointed out that SARS-CoV-2 leads to lymphocyte depletion and inhibits immune function, which is a potential immunological mechanism for the occurrence and progression of COVID-19 [177]. Moreover, lymphocyte counts in blood is also a viable and accurate index to classify the severity (moderate, severe and critical) of COVID-19 patients [12]. Therefore, lymphopenia could be a clinical indicator for ranking the severity of COVID-19 patients. Similarly, following infections of FIPV [178] and RHDV [179], lymphopenia is also related to the severity of disease. A previous study indicated that lymphopenia was related to disease severity in FIPV infection, and the absolute count of lymphocytes in peripheral blood was recognized as the predictor of the disease outcome [178]. In addition, severe lymphocytes depletion was observed in rabbits infected with RHDV, especially at 6 h before death of the infected rabbits [99]. Although it is not clear whether lymphopenia promotes the progression of the disease or the severe disease contributes to lymphopenia, it determines that lymphopenia is associated with the severity of the disease. Lymphopenia also can be served as an important indicator for dynamic assessment of disease status.

### 3.2. Increases in Opportunistic Infection

Patients with immunosuppressive conditions are prone to have opportunistic infection, which could be either severe or more frequent. Opportunistic infection is a significant feature during the HIV infection, which could result in high morbidity and even mortality [180,181]. Lymphopenia is likely to delay viral clearance, in favor of macrophage stimulation and the accompanying “cytokine storm”, which results in the dysfunction of host organs [14,182]. These damages could increase the risk of developing opportunistic infections. A report described that lymphopenia in COVID-19 patients could increase the risk of developing opportunistic infections of mucormycosis, while the recovery of lymphocytes count could improve the acquired immune response and induce the production of mucorales-specific T cells [183]. To the best of our knowledge, there are few reports of opportunistic infections in specific viral diseases accompanied by lymphopenia. Therefore, further investigations are required to reveal the relationship between opportunistic infection and lymphopenia.

## 4. Conclusions

Lymphocytes, especially specific T cells, have a critical role in viral clearance. Thus, lymphopenia may affect the host adaptive immune responses and impact the clinical course of acute viral infections. In this review, we found that lymphopenia was often seen in patients and animals infected with viruses that could result in serious illness or even death and it was found that lymphopenia is associated with disease severity. Seven different mechanisms were involved in lymphopenia caused by viral infections, including cell death, elevated cytokines, chemokines and growth factors, inhibition of lymphopoiesis, lymphocyte trafficking, up-regulated expression of co-inhibitory molecules, metabolic disorders and elevated glucocorticoids. As it has been discussed in this review, lymphopenia could be caused by different viral infections through multiple mechanisms mentioned above or depending on a certain mechanism. Although the potential mechanisms have been widely reported, the molecular mechanism of these pathways still remains poorly understood, which needs to be addressed in the future.

## Figures and Tables

**Figure 1 viruses-13-01876-f001:**
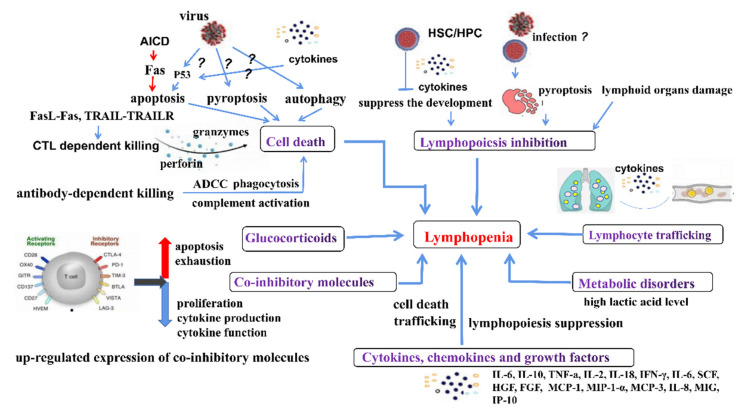
The underlying mechanisms of lymphopenia caused by viral infections.

**Table 1 viruses-13-01876-t001:** Lymphopenia caused by viral infections and the possible mechanisms involved.

Family		Species	Lymphocyte Subsets	Percentage Cases	Possible Mechanisms	Reference
RNA	*Coronaviridae*	SARS-CoV-2	CD8+ T cells, CD4+ T cells, B cells, NK cells	85% of the severe COVID-19 patients	cell death cytokines lymphopoiesis trafficking co-inhibitory molecules metabolic disorders	[5,14,15,16,17,18,19,20,21,22]
		Middle east respiratory syndrome coronavirus (MERS-CoV)	T cells	34% of the MERS-CoV patients	cell death cytokines lymphopoiesis trafficking	[23,24,25,26]
		Severe acute respiratory syndrome coronavirus (SARS-CoV)	CD4+ T cells, CD8+ T cells, B cells, NK cells	90–100% patients in the acute phase of SARS patients	cell death cytokines lymphopoiesis trafficking glucocorticoids	[27,28,29,30,31,32]
		Feline infectious peritonitis virus (FIPV)	CD4+ T cells, CD8+ T cells, B cells, NK cells	77% of FIPV infected-cats	cell death cytokines	[33,34,35,36]
		Canine coronavirus (CCoV)	Lymphocytes	unknown	unknown	[37]
		Equine coronavirus (ECoV)	Lymphocytes	81% of ECoV infected-horses	unknown	[38]
	*Retroviridae*	HIV	CD4+ T cells	49.17–65.2% of AIDS patients	cell death cytokines lymphopoiesis trafficking co-inhibitory molecules	[39,40,41,42,43,44,45,46,47]
		Simian immunodeficiency virus (SIV)	CD4+ T cells	unknown	cell death trafficking	[48,49,50]
		Bovine leukemia virus (BLV)	B cells	unknown	unknown	[51]
	*Filoviridae*	EBOV	CD4+ T cells, CD8+ T cells, NK cells	unknown	cell death cytokines trafficking co-inhibitory molecules	[13,52,53,54,55,56]
		Marburg virus (MARV)	T cells, B cells, NK cells	unknown	cytokines metabolic disorders	[57,58,59]
		Bundibugyo virus (BDBV)	Lymphocytes	unknown	cytokines	[60]
	*Orthomyxoviridae*	Influenza A virus (IAV), IAV H5N1 IAV H7N9 IAV H1N1 IAV H5N6	CD4+ T cells, CD8+ T cells, B cells, NK cells	unknown	cell death cytokines lymphopoiesis trafficking co-inhibitory molecules	[61,62,63,64,65,66]
	*Paramyxoviridae*	Measles virus (MV)	CD4+ T cells, CD8+ T cells, B cells	unknown	cell death lymphopoiesis	[67,68]
		Human parainfluenza virus type 3 (HPIV3)	CD4+ T cells, CD8+ T cells, B cells	unknown	unknown	[69]
		Canine distemper virus (CDV)	CD4+ T cells, CD8+ T cells, B cells	unknown	cell death lymphopoiesis	[70,71,72]
		Peste des petits ruminants virus (PPRV)	Lymphocytes	unknown	unknown	[73,74]
	*Pneumoviridae*	Respiratory syncytial virus (RSV)	CD4+ T cells, CD8+ T cells	unknown	cell death cytokines co-inhibitory molecules metabolic disorders	[75,76,77,78]
	*Arteriviridae*	Porcine reproductive and respiratory syndrome virus (PRRSV)	T cells, B cells	unknown	cell death cytokines lymphopoiesis	[79,80]
	*Picornaviridae*	Foot-and-mouth disease virus (FMDV)	CD4+ T cells, CD8+ T cells, B cells	unknown	cell death cytokines trafficking	[81,82,83,84,85]
		Seneca Valley virus (SVV)	B cells	unknown	cell death	[86]
	*Flaviviridae*	Classical swine fever virus (CSFV)	CD4+ T cells, CD8+ T cells, B cells	unknown	cell death cytokines lymphopoiesis co-inhibitory molecules	[87,88,89,90]
		Hepatitis C virus (HCV)	T cells	6% of HCV-infected patients with acute exacerbation	cell death co-inhibitory molecules	[91,92]
		Border disease virus (BDV)	Lymphocytes	unknown	unknown	[93]
		Dengue Virus (DENV)	Lymphocytes	91.7% of DENV-infected patients	unknown	[94]
		West Nile virus (WNV)	Lymphocytes	unknown	unknown	[95]
		Bovine viral diarrhoea virus (BVDV)	CD4+ T cells, CD8+ T cells,	unknown	cell death lymphopoiesis co-inhibitory molecules	[96]
	*Caliciviridae*	Human norovirus (HNoV)	T cells, B cells	unknown	cytokines trafficking	[97,98]
		Rabbit haemorrhagic disease virus (RHDV)	Lymphocytes	unknown	cell death	[99]
		Feline calicivirus (FCV)	Lymphocytes	unknown	unknown	[100,101]
	*Arenaviridae*	Lymphocytic choriomeningitis virus (LCMV)	T cells, NK cells	unknown	unknown	[102]
	*Togaviridae*	Chikungunya virus (CHIKV)	Lymphocytes	unknown	unknown	[103]
DNA	*Asfarviridae*	African swine fever virus (ASFV)	Lymphocytes	unknown	cell death	[104]
	*Circoviridae*	Porcine circovirus type 2 (PCV-2)	CD3+CD4+CD8+ T cells, CD3+CD4+CD8- T cells, CD3+CD4-CD8+ T cells, CD3+CD4-CD8- T cells, B cells, NK cells	unknown	unknown	[105]
	*Herpesviridae*	Cytomegalovirus (CMV)	CD4+ T cells, CD8+ T cells, NK cells	unknown	unknown	[106]
		Marek’s disease virus (MDV)	B cells	unknown	unknown	[107]
	*Alphaherpesvirinae*	Herpes simplex virus (HSV)	Lymphocytes	unknown	unknown	[108]
	*Parvoviridae*	Feline parvovirus (FPV)	Lymphocytes	unknown	unknown	[109]
		Canine parvovirus (CPV)	Lymphocytes	unknown	unknown	[110]

## Data Availability

Not applicable.

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
