# Peer review of "Lymphopenia Caused by Virus Infections and the Mechanisms Beyond"

_viruses, 2021, doi:10.3390/v13091876_

Round 1

Reviewer 1 Report

The review by Zijing Guo et al. is focused on a very critical phenomena-lymphopenia  in host caused by virus infection.  The manuscript is for the most part well-written and interesting. Some suggestions are listed below.

Major concerns

  1. In the introduction part, the author mentioned the causes of lymphopenia. Sepsis is a devastating illness , and clinical studies have demonstrated that circulating levels of lymphocytes was significantly reduced during the onset of sepsis and can remain depressed for up to 28 days. So obvious reduction in blood lymphocytes count occurs also due to sepsis.
  2. In the second part of this review, Occurrence of lymphopenia during viral infections, in the article by  Canelli E, Catella A, Borghetti P, et al. Phenotypic characterization of a highly pathogenic Italian porcine reproductive and respiratory syndrome virus (PRRSV) type 1 subtype 1 isolate in experimentally infected pigs. Vet Microbiol. 2017;210:124-133. doi:10.1016/j.vetmic.2017.09.002.  This article mentioned porcine reproductive and respiratory syndrome virus (PRRSV) could cause  severe lymphocytopenia in bronchial lymph-nodes and thymus. And PRRSV is classified in the Arteriviridae family within the order  It contains a single-stranded positive-sense RNA of approximately 15 kb in length that encodes at least 9 open reading frames (ORF). Please included this RNA virus in the table 1.
  3. In mechanisms of lymphopenia associated with viral infections part, the interaction or competition between granulopoiesis and lymphopoiesis in the bone marrow was not well illustrated. Please include how those hyperproinflammatory cytokines produce by activated macrophages, neutrophils and monocytes, dendritic cells, endothelial and epithelial by virus that affect the process of lymphopoiesis in the bone marrow .
  4. The authors didn’t mention the common form of lymphopenia like Thymus suppression mediated lymphopenia, Activation induced cell death (AICD) of lymphocytes, Dendritic cell (DC)-dependent killing of lymphocytes which are common in viral infection. It would be fruitful if they mentioned these things.
  5. In Figure 1. the mechanisms picture is not clear and some font is too small. Also the spelling of apoptosis is not right.
  6. As there are a large number of abbreviations word in this review. Please include a list of all abbreviations for this paper.
  7. Some minors spelling issue, in table 1, some place the cell death was “cell death1” , and line 49 “COVID-19”.

Reviewer 2 Report

Lymphocytes have specific roles in the immune response against viral infections as either effector cells or regulator cells. This review has summarized the virus-induced lymphopenia and its potential mechanisms. The following suggestions should be followed for the revision.

1. In this review, authors described that the virus-induced lymphopenia is “widely reported” and lots of viruses can result in lymphopenia. As a matter of fact, lymphopenia is not a universal finding in viral infections except severe infections. Generally, most viruses lead to relative lymphocytosis, while only a few viruses, such as HIV and influenza result in lymphopenia. Therefore, the information provided in this manuscript might mislead the audience. In order to avoid this kind of misleading, the authors should mention that lymphopenia is not a universal finding in viral infections and give a certain percentage of lymphopenia cases in each virus-infected cases listed in table one.

2. As lymphocytes consist of T cells, B cells, and natural killer cells. Lymphopenia can be identified by the type of lymphocyte affected. For example, HIV specifically targets CD4 T cells for infection, resulting in massive losses of that specific cells. Therefore, if the table one can give the data of type of lymphocytes affected, it would be much helpful.

3. Since the table one gives a detail of lymphopenia caused by each virus, the text description of these viruses in section 2, “occurrence of lymphopenia during viral infections” is not necessary.

4. The manuscript suffers from grammatical and editorial errors, some of which are listed below.  The manuscript should be re-read carefully and checked for all the errors.

(1) Plural nouns should be used. For examples, “by virus infection” at line 2 and “due to virus infection” at line 27, should be replaced with “by virus infections” at line 2 and “due to virus infections” at line 27.

(2) Wrong tenses were used. For example, “lymphopenia was often seen in patients” at line 109.

(3) Subject-verb disagreement, for examples, “different viruses seems” at line 116 and “the synergic action of CCGFs play” at line 206, etc.

Round 2

Reviewer 1 Report

Overall the revised version answered most of my previous concerns.

Two suggestions are listed below.

  1. In the introduction part, a bunch of reports have proved that lymphopenia could be caused by cancers. Please include this part in the introduction.
  2. In table 1, for lane coronavirus (SARS-CoV), look like some information was missing.
